# Plasma from Patients with Rheumatoid Arthritis Reduces Nitric Oxide Synthesis and Induces Reactive Oxygen Species in A Cell-Based Biosensor

**DOI:** 10.3390/bios9010032

**Published:** 2019-02-27

**Authors:** Herbert Herlitz-Cifuentes, Camila Vejar, Alejandra Flores, Paola Jara, Paulina Bustos, Irene Castro, Evelyn Poblete, Katia Saez, Marina Opazo, Jorge Gajardo, Claudio Aguayo, Estefania Nova-Lamperti, Liliana Lamperti

**Affiliations:** 1Department of Clinical Biochemistry and Immunology, Faculty of Pharmacy, Universidad de Concepción, Victor Lamas 1290, Concepcion 4030000, Chile; herbert.herlitz@uss.cl (H.H.-C.); camilavejar@udec.cl (C.V.); daniale.floresf@gmail.com (A.F.); paojara@udec.cl (P.J.); pbustos@udec.cl (P.B.); epoblete1@uc.cl (E.P.); caguayo@udec.cl (C.A.); llampert@udec.cl (L.L.); 2Facultad de Ciencias de la Salud, Universidad San Sebastián, Lientur 1457, Concepcion 4030000, Chile; 3Clinical and Genetic Laboratory PreveGen, Chacabuco 556, Concepcion 4030000, Chile; 4Department of Rheumatology, Regional Clinical Hospital Dr. Guillermo Grant Benavente, San Martin 1436, Concepcion 4030000, Chile; ircastro@udec.cl; 5Department of Internal Medicine. Faculty of Medicine, Universidad de Concepción, Victor Lamas 1290, Concepcion 4030000, Chile; jgncardio@gmail.com; 6Department of Statistics, Faculty of Physical Sciences and Mathematics, Universidad de Concepción, Victor Lamas 1290, Concepcion 4030000, Chile; ksaez@udec.cl; 7Central Clinical Laboratory, Regional Clinical Hospital Dr. Guillermo Grant Benavente, San Martin 1436, Concepcion 4030000, Chile; mopazo@ssconcepcion.cl

**Keywords:** rheumatoid arthritis, cardiovascular (CV) diseases, endothelial cell-oxidative stress

## Abstract

Rheumatoid arthritis (RA) has been associated with a higher risk of developing cardiovascular (CV) diseases. It has been proposed that systemic inflammation plays a key role in premature atherosclerosis development, and is therefore crucial to determine whether systemic components from RA patients promotes endothelial cell-oxidative stress by affecting reactive oxygen species (ROS) and nitric-oxide (NO) production. The aim of this study was to evaluate whether plasma from RA patients impair NO synthesis and ROS production by using the cell-line ECV-304 as a biosensor. NO synthesis and ROS production were measured in cells incubated with plasma from 73 RA patients and 52 healthy volunteers by fluorimetry. In addition, traditional CV risk factors, inflammatory molecules and disease activity parameters were measured. Cells incubated with plasma from RA patients exhibited reduced NO synthesis and increased ROS production compared to healthy volunteers. Furthermore, the imbalance between NO synthesis and ROS generation in RA patients was not associated with traditional CV risk factors. Our data suggest that ECV-304 cells can be used as a biosensor of systemic inflammation-induced endothelial cell-oxidative stress. We propose that both NO and ROS production are potential biomarkers aimed at improving the current assessment of CV risk in RA.

## 1. Introduction

Rheumatoid arthritis (RA) is a systemic autoimmune disease of unknown etiology characterized by chronic inflammation of synovial membranes and articular structures of multiple joints [1,2]. RA causes pain, impaired function of the joints, disability and concomitant comorbidities, particularly in the cardiovascular system [2]. Cardiovascular disease (CVD) is the most common comorbidity in RA and the prevalence of CVD in patients with RA is more than doubled compared with the general population [3]. Moreover, CVD is the leading cause of death in patients with RA [4], with a standardized mortality ratio of 1.5 and a 50% risk of death [5]. In fact, the magnitude of risk of CVD in RA has been compared to type-2 diabetes [6], and several studies indicate that RA is an independent risk factor for myocardial infarction or stroke [7,8]. Specific causes of CV events in RA remain unclear [9]. An accelerated or premature atherosclerosis due to systemic inflammation has been proposed as the main cause. Circulating cytokines alter the function of adipose tissue, skeletal muscle, liver cells and vascular endothelium, promoting a spectrum of CV risk factors, including insulin resistance, dyslipidaemia, oxidative stress and endothelial dysfunction [10]. However, these traditional cardiovascular or Framingham risk factors seem to be inadequate in predicting coronary atherosclerosis in patients with RA [11]. In fact, these factors have not been differentially increased in RA patients compared to healthy controls [12,13].

Several studies have reported the presence of early microvascular [14,15,16] and macrovascular [17] endothelial dysfunction in RA patients. In addition, several studies have demonstrated the presence of early subclinical atherosclerosis in RA patients with no previous medical history of CVD [17,18,19]. Furthermore, it has been shown that early diagnosis of endothelial dysfunction correlates with RA disease activity [20]. Therefore, biomarkers of endothelial damage and systemic inflammation may be relevant for the prediction of early cardiovascular events in patients with RA. 

Intracellular reactive oxygen species (ROS) production and Nitric Oxide (NO) synthesis are two endothelial cell responses that need to be balanced in order to maintain vascular homeostasis. Whereas NO is a vasoactive factor from the vascular endothelium aim at maintaining an atheroprotective environment [10,20], ROS are associated with injury and endothelial cell activation, thus increased levels of ROS and decreased levels of NO indicate endothelial cell-oxidative stress. This persistent oxidative stress may cause vascular damage, promoting endothelial dysfunction and CVD. Since systemic ROS and NO measurements are not representative of endothelial oxidative stress, it is ideal to use a biosensor that responds like an endothelial cell to mimic the endothelial stress potentially induced by systemic factors from the plasma of RA patients. The aim of this study was to evaluate the capacity of plasma from RA patients to promote ROS production and impair NO synthesis by using ECV-304 cells as an endothelial-like biosensor.

## 2. Materials and Methods

### 2.1. Study Protocol

Seventy-three patients with RA from the Regional Hospital in Concepcion, fulfilling the criteria of the American College of Rheumatology for RA were enrolled in this study. As a control group, fifty-two age-matched healthy volunteers were recruited from the University of Concepción. All patients were taking disease modifying anti-rheumatic drugs (DMARDs) at the time of enrolment. Exclusion criteria for RA patients: other autoimmune disease, other inflammatory disease, seronegative arthritis and previous history of coronary heart disease or cerebrovascular accident. Exclusion criteria for healthy volunteers: pregnancy, recent infectious or inflammatory diseases and previous history of coronary heart disease or cerebrovascular accident. All participants gave written informed consent prior enrolment, and samples were processed and analyzed in a blinded fashion. This study was approved by the ethics committee of Concepcion Health Service and University of Concepcion-Chile and our research was conducted in accordance with the Declaration of Helsinki (1964).

Body Mass Index (BMI), waist circumference, blood pressure, hypertension status, diabetes status, smoking status and family history of coronary-artery disease were recorded from all participants. In patients, disease activity was measured by using the Disease Activity Score based on the evaluation of 28 joints (DAS28). This score was calculated using the number of swollen and tender joints (28 joint count), the patient global assessment, the medical global assessment and the erythrocyte sedimentation rate (ESR). The ability to perform activities of daily living was also evaluated by using a health assessment questionnaire (HAQ).

### 2.2. Sample Collection

Blood samples from patients and healthy controls were obtained after overnight fasting. Plasma was collected after centrifugation and stored at −80 °C. Insulin, glucose, total cholesterol, high-density lipoprotein (HDL) cholesterol, low-density lipoprotein (LDL) cholesterol and triglycerides were measured in all samples. The ESR, anti-cyclic citrullinated peptide antibodies (anti-CCP) and rheumatoid factor (RF) were measured only in RA patient samples. 

### 2.3. ECV-304 Cell Culture

ECV-304 cells were cultured at 37 °C 5% CO_2_ in 199 medium (Gibco) containing 10% (v/v) foetal calf serum (Thermo Scientific, HyClone Laboratories Inc) and 100 U/mL penicillin-streptomycin. 2 × 10^4^ cells/100 μL/well ECV-304 cells were added in 96-well plates. After a uniform formation of a single layer of endothelial cells was confirmed, the original medium was removed, and the cells were incubated for 12 h with plasma (100 μL/well) from patients with RA or healthy control in triplicate. The ECV-304 cell line was provided by Dr. Juan Carlos Vera.

### 2.4. Intracellular ROS Production

ROS formation was determined by the method used by Searle et al. [21], using the probe 2,7-dichlorofluorescein diacetate (DCF) (Calbiochem). ECV-304 cells were cultured in 96 well plates before plasma addition. After 12 h of plasma incubation or positive control H_2_O_2_ [0.1 mM and 1 mM], ECV-304 cells were washed with PBS and cultured with 2.5 μM DCF for 30 min at 37 °C 5% CO_2_ in 199 media. Fluorescence intensity was then measured using a microplate reader at emission 540 nm (excitation 488 nm) (Sinergy 2, Biotek). The basal fluorescence intensity was measured prior H_2_O_2_ [10 mM] stimulation. Then, the plate was activated with H_2_O_2_ [10 mM] and the fluorescence signal of intracellular ROS production was measured at 60, 180, 300 and 600 s. Fluorescence intensity from H_2_O_2_-stimulated ECV-304 cells in the absence of DCF was used as a negative control. Fluorescence intensity per sample was normalized by total protein concentration on each well. After obtaining the IF/ug per sample, fluorescence intensity at 0 s (basal level) was subtracted from fluorescence intensity at 0, 60, 180, 300 and 600 s post-activation with H_2_O_2_ [10 uM] and tabulated to calculate the kinetics parameters of ROS production. Maximum fluorescence intensity (Fmax) and Km for each sample was obtained according to the Michaelis-Menten equation, using a Graphpad prism (Graphpad prism software, San Diego, CA 92108, USA.). Final values were obtained by dividing the maximum fluorescence intensity (Fmax) by Km (t_1/2_). (Scheme 1). Cut off values to define patients with high ROS were calculated based on ROC curves.

### 2.5. Intracellular Synthesis of NO

NO was determined accordingly by using the method previously described [22]. ECV-304 cells were cultured in 96 well plates before plasma addition. After 12 h of plasma incubation, ECV-304 cells were washed with PBS and incubated with PBS containing 4,5-diaminofluorescein diacetate (DAF-2DA, 5 μM, 30 min, 37 °C, 5% CO_2_) (Calbiochem). Fluorescence intensity was then measured using a microplate reader at emission 540 nm (excitation 488 nm) (Sinergy 2, Biotek). The basal fluorescence intensity was measured before Histamine [10 mM] addition. Then, the plate was activated with Histamine [10 mM] and the fluorescence signal of NO was measured at 60, 180, 300 and 600 s. Fluorescence intensity from Histamine-stimulated ECV-304 cells in the absence of DAF-2DA was used as a negative control. In addition, the nitric oxide synthase inhibitor N-nitro l-arginine methyl ester (l-NAME) was added 30 min before Histamine activation to confirm NO synthesis. Fluorescence intensity per sample was normalized by total protein concentration on each well. After obtaining the IF/ug per sample, fluorescence intensity at 0 s (basal level) was subtracted from fluorescence intensity at 0, 60, 180, 300 and 600 s post-activation with Histamine [10 uM] and tabulated to calculate the kinetics parameters of NO synthesis. Maximum fluorescence intensity (Fmax) and Km for each sample was obtained according to Michaelis-Menten equation, using Graphpad prism (Graphpad prism software, San Diego, CA 92108, USA). The slope of NO production per sample was obtained by dividing the maximum fluorescence intensity (Fmax) by Km (t_1/2_) (Scheme 2). Cut off values to define patients with low NO were calculated based on ROC curves. 

### 2.6. IL-6, sVCAM-1 and hs-CRP Levels in Plasma

IL-6 and sVCAM-1 levels in plasma were measured using ELISA according to manufacturer’s instructions (eBioscience ELISA Ready-SET-Go). Concentrations of hs-CRP were quantified using a chemiluminescent immunometric solid phase assay (Siemmens).

### 2.7. Statistical Analysis

A comparison between RA patients and healthy controls for all parameters was performed. The results are presented as mean ± standard deviation (SD). Normality was verified using the Shapiro-Wilk test. Logarithmic transformation was used in variables with positive bias. Non-parametric Mann-Whitney test and Kruskal-Wallis test were used for those variables that did not fulfil the assumption of normality test. Non-normally distributed data were presented as median and interquartile ranges. All statistical procedures were analyzed with SPSS Statistics and represented with Graphpad prism (Graphpad prism software Inc.,).

## 3. Results

### 3.1. RA Patients and Healthy Control Exhibit Similar Traditional CV Risk Factors

The clinical parameters of patients with RA are described in Table 1. Patients exhibited mean disease duration of 14.5 ± 9.7 years, with a moderate disease activity (DAS28 = 3.7 ± 1.2). Their disability was also moderate, and the mean score of the HAQ survey was 1.3 ± 0.9, indicating a moderate self-perceived level of physical disability. In addition, 19 patients (26.8%) were positive for anti-CCP and 45 patients (62.5%) were seropositive for RF. All patients were under DMARD treatment, mainly in combination with corticosteroids (99%). Metabolic parameters of patients with RA and healthy controls are described in Table 2. No significant differences between RA patients and healthy controls were observed when age, weight, height and BMI were compared. However, the waist circumference was found significantly increased in RA patients compared to healthy controls. In fact, the prevalence of central obesity (waist circumference ≥ 88cm in women) was 81% in RA patients compared to 56% in healthy controls (*P* = 0.0029). Moreover, the systolic and diastolic blood pressure was increased in patients with RA (*P* < 0.01), and a higher prevalence of hypertension (47% vs. 23%, *P* < 0.01) was also observed in this group. The prevalence of family history of heart disease was higher in healthy controls than in RA patients (23% vs. 7%, *P* < 0.01). Although both groups had elevated concentrations of total cholesterol, the control group had significantly higher levels of LDL cholesterol than RA patients. No significant differences in the concentration of triglycerides, HDL cholesterol, insulin and HOMA-IR were found between groups. According to the median HOMA-IR, both groups had borderline values for insulin resistance, using the defined cut-off values for the insulin resistance HOMA1 formula ≥ 2.5 [23].

### 3.2. ECV-304 Cells Incubated with Plasma from RA Patients Exhibited Reduced NO Synthesis and Increased ROS Production Compared To Healthy Volunteers 

In order to evaluate the potential injurious effect of plasma from RA patients in comparison with plasma from healthy volunteers, ECV-304 cells were incubated with plasma from both groups for 12 h. This cell line was used as a biosensor to determine whether plasma from patients with systemic inflammation promotes ROS production and/or impairs NO synthesis. The maximum fluorescence for each probe was obtained incubating ECV-304 cells with 10 mM H_2_O_2_ for ROS production and 10 μM histamine for NO synthesis. ECV-304 cells incubated with 10 mM H_2_O_2_ or 10 μM histamine in the absence of the fluorescent probe were used as a negative control. Culture medium from non-stimulated ECV-304 cells in the presence of the fluorescent probe was used to obtain the baseline fluorescence for NO synthesis and ROS production (Figure 1a).

Figure 1b shows that ROS production was significantly higher in ECV-304 cell cultures exposed to plasma of patients with RA compared to healthy controls (**** *P* < 0.0001) (Figure 1b). Concomitantly, Figure 1c shows that NO synthesis was significantly reduced in ECV-304 cell cultures exposed to plasma of patients with RA compared to controls (**** *P* < 0.0001) (Figure 1c). These results indicate that plasma components from RA patients induce endothelial cell-oxidative stress and impair endothelium-dependent vasodilation compared to healthy subjects. Finally, we evaluated non-traditional cardiovascular risk factors associated with inflammation (hs-CRP and IL-6) and an endothelial activation biomarker (sVCAM-1) in plasma samples from RA patients and healthy controls (Figure 1d). The concentration of hs-CRP, IL-6 and sVCAM-1 were significantly increased in plasma from RA patients compared to healthy controls (Figure 1d), indicating that RA patients had a higher prevalence of presenting non-traditional cardiovascular risk factors. Moreover, RA patients had a higher prevalence of presenting vascular damage considering the results of sVCAM-1.

### 3.3. Endothelial Cell Oxidative Stress in RA Patients is Not Associated with Traditional CV Risk Factors 

Having shown the significant difference between RA patients and healthy controls in ROS and NO synthesis, we defined healthy controls and RA patients with high or normal ROS production, and with low or normal NO production based on the interpretation of ROC curves obtained from all donors. For ROS production, our cut off was 1.66 IF/ug, whereas for NO our cut off was 0.042 IF/ug. When data from individuals with differential ROS production was analyzed, we observed higher concentrations of IL-6 and sVCAM-1 in plasma from RA patients with high ROS in comparison with plasma from healthy controls with normal or high ROS. However, these differences were not statistically significant between RA patients with normal or high ROS (Figure 2). No significant differences were observed when traditional risk factors were compared between groups, except for Glucose levels that were decreased in RA patients with high ROS (Figure 2). Similar results were observed when NO was evaluated (Figure 3). These results suggest that RA patients with high ROS or low NO, exhibit higher systemic inflammation in comparison with individuals without RA, regardless their oxidative stress state. However, inflammatory parameters did not correlate with ROS or NO synthesis in RA patients (Data not shown). Finally, parameters associated with disease activity were not statistically significant between sub-groups of patients with RA, suggesting that biomarkers aimed at detecting early vascular damage, endothelial cell-oxidative stress or systemic inflammation, could improve the current assessment of CV disease in patients with RA (Table 3).

## 4. Discussion

Cardiovascular disease is the leading cause of death in patients with RA [3,4,24]. However, traditional CVD risk factors such as hypertension, hyperlipidaemia, diabetes and smoking, not always correlate with the development of CVD in this autoimmune disorder [12]. As it has been proposed that systemic inflammation plays a role in the development of endothelial cell damage, we evaluated the capacity of plasma from RA patients to promote ROS production and impair NO synthesis by using ECV-304 cells as a biosensor of endothelial cell oxidative stress. Higher oxidative responses in cells cultured with plasma from patients with RA in comparison with plasma from healthy volunteers were observed. Interestingly, similar traditional CVD risk factors were present in both groups. For example, the prevalence of diabetes and smoking was not significantly different between groups, whereas previous history of CVD in first-degree relatives was more prevalent in the control group. Similarly, despite no difference in the lipid profile between RA patients and healthy volunteers, LDL cholesterol was also significantly higher in the control group. Hypertension, on the other hand, was significantly increased in RA patients compared to volunteers, as previously shown [12,13]. This indicates that patients with RA did not exhibit significant differences in terms of traditional CV risk factors in comparison with our control group, suggesting that other factors should be included in order to improve the assessment of CV risk in patients with RA.

ROS production and NO synthesis measurements in the endothelial cell line ECV-304 exposed to plasma from patients with RA have not been reported to date. We have demonstrated for the first time that plasma from patients with RA is injurious to endothelial cells. Hence, plasma from patients with RA induces endothelial oxidative stress, by deregulating ROS production and NO synthesis. Our research group has previously reported a similar system using human umbilical vein endothelial cells incubated with serum from hypercholesteraemic patients, observing reduced NO synthesis and increased ROS formation in patients compared with healthy controls [21]. Both studies support the use of cells as a biosensor of oxidative stress in response to systemic factors, however the use of human umbilical vein endothelial cells present higher donor variability, a time-consuming process to get enough cells to perform experiments and more expensive resources for isolation and cell culture. Thus, ECV-304 cells represent a better candidate as a biosensor to analyze oxidative stress induced by systemic factors. Other studies have analyzed oxidative stress systemically, for example studies from Mateen et al. have compared the antioxidant status between RA patients and healthy controls by monitoring ROS production and oxidative damage markers in periphery. Their results showed higher oxidative stress in RA patients compared to healthy controls, and a positive correlation between ROS production and lipid peroxidation, protein oxidation and DNA damage in RA patients [25]. Interestingly, no significant difference in ROS production between seropositive and seronegative RA patients was observed [25]. In a further study, same authors evaluated whether the presence of inflammatory cytokines correlated with ROS production. Results revealed that levels of cytokines, such as TNF-α, IL-1β, IL-17, IL-10 and IL-6, had a positive and significant correlation with levels of ROS [26]. They also found a positive correlation between ROS production and NO synthesis in the plasma, whereas we found reduced NO synthesis induction by plasma from RA patients in comparison with control plasma. The discrepancy between the studies could be associated with the fact that we measured both parameters in cells, whereas they measured ROS in the haematocrit and NO in the plasma. 

The association between systemic inflammation and NO synthesis in RA has also been evaluated as it is well known that a constitutive production of NO is required to maintain an atheroprotective environment within the vascular system. In this context, evaluation of the symmetric (SDMA) and asymmetric (ADMA) dimethylarginine has become an indicator of NO synthesis, as these analogues of L-arginine are endogenous inhibitor of the NO synthase because they compete with L-arginine at the active site of the enzyme. This indicates that high ADMA levels results in low NO synthase activity, and consequently reduced NO synthesis. Recent studies have analyzed the association between inflammatory parameters, SDMA/ADMA levels in plasma and cardiovascular risk factors. Studies from Kwaśny-Krochin et al. showed higher ADMA levels in patients with RA compared to healthy controls [27]. Moreover, ADMA levels in the RA group correlated positively with CRP and disease activity, but not with age, renal function, or the medications used. They also demonstrated that CRP was the only independent predictor of ADMA levels in RA [27]. Another study showed that CRP had a significant positive correlation with ADMA levels after 6 years of follow up, but not at baseline, suggesting that progressive inflammation may be responsible for the reduction of NO synthesis, independently of classical risk factors [28]. A more recent study has also hypothesized that ADMA levels is associated with endothelial function in RA, therefore carotid intima media thickness (cIMT) and arterial stiffness as well as non-invasive assessments were measured and correlated with ADMA levels. The authors concluded that microvascular function, arterial stiffness and cIMT were associated with circulating ADMA levels in RA patients with high inflammatory markers such as erythrocyte sedimentation rate and CRP [29]. Altogether these results suggest that direct or indirect ROS and NO measurements could be potential biomarkers aimed at identifying whether systemic factors promote endothelial cell-oxidative responses. 

In term of inflammation, all RA patients in this study have active disease and high levels of inflammation as 58% of RA patients had elevated levels of hs-CRP, 40% exhibited high levels of IL-6 and 55% had high levels of sVCAM-1. It is known that systemic inflammation has detrimental effects in several tissues, particularly in the endothelium where stimulates atherosclerosis [30]. Several biomarkers of systemic inflammation have been associated with endothelial dysfunction and atherosclerosis, for example, IL-6 and CRP have been inversely correlated with the thickness of the intima media of the carotid in RA patients [30,31]. Consistent with our results, CRP levels were higher in patients with high ROS production and low NO synthesis than healthy volunteers with normal ROS or NO. Furthermore, the levels of sVCAM-1 and IL-6 were significantly higher in RA patients than healthy volunteers, either with an altered or normal ROS and NO production. This data indicates that both parameters are valuable candidates to analyze endothelial status or non-specific inflammation systemically. From an autoimmune point of view, 62.5% of patients were positive for the Rheumatoid Factor and only 26.8% were positive for the anti-CCP. One study reported a relationship between the presence of these autoantibodies and endothelial function. They showed that positive patients for anti-CCP and RF have an impaired endothelial function, due to a higher hyperaemic reactive index, independent of other CV risk factors [32].

Since the inflammatory state and the autoimmune components are the main differences between the plasma from RA patients and healthy controls, it is possible that some of these components alone or in combination are causing deregulation in ROS and NO intracellular production, as observed in our model of biosensor cells. Previous analyses have shown phenotypic changes in the endothelium in the onset of RA in an experimental model of arthritis induced by adjuvant. The authors identified mechanisms involved in endothelial dysfunction, as well as a correlation between endothelial dysfunction and biomarkers of systemic inflammation [33]. Despite these efforts, the inflammatory mediators involved in endothelial damage have not been fully defined which are [31,33,34]. It would be interesting to elucidate which of the plasma components is causing endothelial cell damage, however, the combination of these different factors may be more relevant than each component alone in the prognosis of these patients. 

Finally, ROS and NO measurements in cell-biosensors, alone or in combination with other vascular parameters could become possible biomarkers for disease activity in RA in terms of endothelial damage. These measurements could help to establish reference values for endothelial cell status or oxidative stress in RA, as ROS production is relatively low in healthy controls. In addition, they could be useful to evaluate therapies aimed at reducing oxidative stress. For example, Khojah et al. have shown that therapy with ascorbic acid (1 g/day) significantly reduced the levels of ROS in RA patients that initially exhibited positive correlations between most of the reactive species and the clinical and biochemical markers of RA. In fact, after treatment with the antioxidant, several parameters were significantly reduced such as DAS28, matrix metalloproteinase 3 and the erythrocyte sedimentation rate [35]. Another study has reviewed the clinical efficacy of dietary antioxidants showing the beneficial role of natural antioxidants in RA and the important role of potential biomarker of oxidative stress for detecting disease pathophysiology in patients with this pathology [36]. These studies suggest that our endothelial biosensor model could help to evaluate potential risk factors individually, as well as potential treatments aimed at reducing endothelial cell-oxidative stress. However, more studies are required to elucidate whether the induction of ROS and NO synthesis by plasma components from patients with RA is associated with endothelial dysfunction and higher incidence of cardiovascular disease in these patients. 

## 5. Conclusions

Our findings demonstrated that ECV-304 cells can be used as a biosensor of oxidative stress induced by plasma components. We demonstrated that plasma from RA patients promote oxidative stress by altering ROS and NO production. Considering the 2015/2016 updated recommendations for the management of CV risk in RA, the inclusion of novel biomarkers of CVD diseases associated with systemic inflammation could improve the assessment of CV risk in patients with RA.

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
