# Peer review of "Plasma from Patients with Rheumatoid Arthritis Reduces Nitric Oxide Synthesis and Induces Reactive Oxygen Species in A Cell-Based Biosensor"

_biosensors, 2019, doi:10.3390/bios9010032_

Reviewer 1 Report

This manuscript described the use of a cell-based biosensor for monitoring NO and ROS production as potential biomarkers for cardiovascular disease risk  in patients affected by Rheumatoid arthritis (RA). In particular the authors evaluated whether plasma from RA patients impair NO synthesis and increase ROS production by using the cell-line ECV-304 as a biosensor.

The manuscript is well written. The experimental plan is rigorous and well design. The number of subjects is adequate for the statistical analysis.

Specific comments:

-Because RA has a higher incidence in females, it is necessary to specify the gender of healthy subjects. Furthermore, an analysis in terms of ROS production and NO synthesis should be done in  function of gender. It is known that antioxidant defense decreases with aging especially in women, being linked to menopause.

-In order to set up and validate the cell-based biosensor, I suggest, beside the negative and positive control (fig.1a), the use of a calibration curve, such as different concentration of H202 or Histamine. This point is important for assessing the reproducibility and reliability of the tool.

-In chapter 3.3 and in Figures 2 and 3, the authors evaluated inflammatory biomarkers, such as IL6, hsPCR and sVCAM1, subdividing sub-groups of controls and RA patients in function of lower or higher ROS or NO values they expressed. In reality, this is not a correlation analysis that would presuppose evaluating the individual values of inflammatory biomarkers vs ROS or NO for each sample. Therefore, I suggest to re-formulate this concept, trying to be more adherent to the meaning of the analysis.

Author Response

Point by Point reply

REVIEWER1

Please see PDF attached.

Reviewer 2 Report

In this manuscript, the author investigated the effect of rheumatoid arthritis patient and healthy donor blood plasma on an endothelial cell line (ECV-304). Changes to intracellular levels of reactive oxygen species and nitric oxide was then evaluated as a potential biosensor for cardiovascular disease. If successful, the author’s approach would be a simple and efficient method to address an important clinical problem. While this work is important, several items must be addressed before that conclusion can be made.

1)     The major issue with the conclusion of the paper is ROS and NO is heavily involved in multiple pathways and diseases. How can this then be used as a biomarker for cardiovascular disease as opposed to other comorbidities of rheumatoid arthritis? The authors should address this in detail.

2)     The method of NO detection was with the use of a fluorescent probe, DCF-2DA. The specificity of this category of probes has been previously questioned [1]. The authors should address this and perhaps include additional detection/validation of NO as a majority of their conclusions are based on this assay.

[1] Roychowdhury, Sanjoy, et al. "Oxidative stress in glial cultures: detection by DAF‐2 fluorescence used as a tool to measure peroxynitrite rather than nitric oxide." Glia 38.2 (2002): 103-114.

3)     On page 3 line 131, the authors write “Results were expressed as maximum fluorescence intensity divided by mean time”. It is unclear what time is referring to here and why it is used in the calculation for results. Is this only referring to the positive and negative controls?

4)     In both Figure 2 and 3, it is unclear what the top right graph is supposed to represent. For Figure 2, it seems that it is graphing ROS vs ROS and Figure 3 is graphing NO vs NO. Additionally, the healthy samples (HC) should also be divided up between normal and high ROS and normal and low NO. This would allow the data to be better interpreted. Do the healthy samples with high ROS also show higher risk factors or is there no difference?

5)     There are several missing words and wording issues throughout the paper. For example, the heading “Plasma levels of IL-6, VSCAM-1, and hs-CRTP” should actually be “IL-6, VSCAM-1, and hs-CRTP levels in Plasma”. These changes will help manuscript read more fluidly.

Author Response

Point by Point reply

Dear REVIEWER 2

Please find attached a PDF file with a Point by Point reply.

Reviewer 3 Report

The study evaluates a cell line for measuring nitric oxide and reactive oxygen species induction by blood plasma, and compare to clinical parameters, especially ones related to cardiovascular disease.

Major issue:

I don't agree with the reasons the authors use to connect NO and ROS to CV disease based on figures 2 and 3. The authors show a difference between controls and both low NO RA and high ROS RA, but not between controls and normal NO or normal ROS, for CV risk factors. They then interpret this as a correlation between ROS level or NO level on one hand and CV risk on the other, even though there is no significant difference between the ROS RA groups or the NO RA groups... it's interpreting absense of evidense as evidense of absense. Do note that P-values do depend on group size (n) a lot, and this size is not equal between groups preventing comparison of P-values. I consider the statement about NO and ROAS a potential biomarkers for CV risk unsupported by their data.

Minor issue:

I cannot see the cutoffs for * and ** written out anywhere.

Originality:

The methods for measuring NO and ROS are not new, nor is measurement of ROS and NO in this cell line (NO: https://www.sciencedirect.com/science/article/pii/S0014579398007789,  ROS: https://www.atsjournals.org/doi/full/10.1165/ajrcmb.20.6.3424), although the intersection between the method and the cell line is new for NO, and the connection to cardiovascular disease appears to be.

Author Response

Point by Point reply

Dear REVIEWER 3

Please find attached a PDF file with a Point by Point reply.

Round  2

Reviewer 1 Report

The authors received all the suggestions referees did. Now the revised manuscript is really improved and for me suitable  for publication.

Author Response

We thank Reviewer1 for the comments in the first round and we agree that all the comments from 3 Reviewers help us to improve the article.

Reviewer 3 Report

I read the reply and I understand the authors' point about originality and the usefulness of a new biosensor, and agree that the study and it's technique has value beyond cardiovascular disease prediction. Apologies for irritation being carried over from the description of figures 2 and 3 (in the previous version of the manuscript).

I like the new version of figure 2 and 3.

The text for figures 2 and 3 is now literally what the figures show. I worry this is still written in a way that can mislead readers, especially with the abstract describing this a as an association between NO/ROS and inflammatory markers. After all, what they show is that both ROS and the inflammatory markers are higher in RA patients (and NO lower). But it ought to be pointed out somewhere that RA patients didn't differ in the inflammatory parameters between the NO-level groups (high or low) or between ROS-level groups. And I wish the abstract didn't highlight that association, as it isn't in their data once case/control status has been controlled for, as they now do in the figures.

Spelling:

Typos on line 331 (CDV) and 226 (asociated), and Michaelis-menten (2 places) is usually spelled Michaelis-Menten.

Author Response

REVIEWER 3

1) I read the reply and I understand the authors' point about originality and the usefulness of a new biosensor, and agree that the study and it's technique has value beyond cardiovascular disease prediction. Apologies for irritation being carried over from the description of figures 2 and 3 (in the previous version of the manuscript).

I like the new version of figure 2 and 3.

The text for figures 2 and 3 is now literally what the figures show. I worry this is still written in a way that can mislead readers, especially with the abstract describing this a as an association between NO/ROS and inflammatory markers. After all, what they show is that both ROS and the inflammatory markers are higher in RA patients (and NO lower). But it ought to be pointed out somewhere that RA patients didn't differ in the inflammatory parameters between the NO-level groups (high or low) or between ROS-level groups. And I wish the abstract didn't highlight that association, as it isn't in their data once case/control status has been controlled for, as they now do in the figures.

We thank Reviewer’s suggestions and in order to avoid misunderstandings in the Abstract, we remove the phrase:

Furthermore, the imbalance between NO synthesis and ROS generation in RA patients was significanlty associated with inflamatory parameters”

Now:

“Furthermore, the imbalance between NO synthesis and ROS generation in RA patients was not associated with traditional CV risk factors”

We now understand that the main concern from Reviewer 3 was that there is no difference between RA patients with high ROS vs normal ROS, or low NO vs normal NO. We were focusing in comparing HC vs RA, therefore we did not get this point the first time. To clarify this result, we have added the following phase in the description of Figure 2 and 3:

When data from individuals with differential ROS production was analyzed, we observed higher concentrations of IL-6 and sVCAM-1 in plasma from RA patients with high ROS in comparison with plasma from healthy controls with normal or high ROS. However, these differences were not statistically significant between RA patients with normal or high ROS (Figure 2).

Spelling:

Typos on line 331 (CDV) and 226 (asociated), and Michaelis-menten (2 places) is usually spelled Michaelis-Menten.

We have changed CDV for CVD.
We have changed associated for associated
We have changed Michaelis-menten for Michaelis-Menten.
